# Patterns of Gene Expression, Splicing, and Allele-Specific Expression Vary among Macular Tissues and Clinical Stages of Age-Related Macular Degeneration

**DOI:** 10.3390/cells12232668

**Published:** 2023-11-21

**Authors:** Treefa Shwani, Charles Zhang, Leah A. Owen, Akbar Shakoor, Albert T. Vitale, John H. Lillvis, Julie L. Barr, Parker Cromwell, Robert Finley, Nadine Husami, Elizabeth Au, Rylee A. Zavala, Elijah C. Graves, Sarah X. Zhang, Michael H. Farkas, David A. Ammar, Karen M. Allison, Amany Tawfik, Richard M. Sherva, Mingyao Li, Dwight Stambolian, Ivana K. Kim, Lindsay A. Farrer, Margaret M. DeAngelis

**Affiliations:** 1Department of Ophthalmology, Ross Eye Institute, Jacobs School of Medicine and Biomedical Sciences, State University of New York, University at Buffalo, Buffalo, NY 14203, USA; treefash@buffalo.edu (T.S.); czhang62@buffalo.edu (C.Z.); leah.owen@hsc.utah.edu (L.A.O.); jhlillvi@buffalo.edu (J.H.L.); jbarr2@buffalo.edu (J.L.B.); parkercr@buffalo.edu (P.C.); robert.finley1@hsc.wvu.edu (R.F.); nadinehu@buffalo.edu (N.H.); elizabethdgeorge@gmail.com (E.A.); ryleezav@buffalo.edu (R.A.Z.); gravee1@g.ucla.edu (E.C.G.); xzhang38@buffalo.edu (S.X.Z.); mhfarkas@buffalo.edu (M.H.F.); 2Neuroscience Graduate Program, Jacobs School of Medicine and Biomedical Sciences, State University of New York, University at Buffalo, Buffalo, NY 14203, USA; 3Department of Ophthalmology and Visual Sciences, University of Utah School of Medicine, The University of Utah, Salt Lake City, UT 84132, USA; akbar.shakoor@hsc.utah.edu (A.S.); albert.vitale@hsc.utah.edu (A.T.V.); 4Department of Population Health Sciences, University of Utah School of Medicine, The University of Utah, Salt Lake City, UT 84132, USA; 5Department of Obstetrics and Gynecology, University of Utah School of Medicine, The University of Utah, Salt Lake City, UT 84132, USA; 6Veterans Administration Western New York Healthcare System, Buffalo, NY 14212, USA; 7Department of Biochemistry, Jacobs School of Medicine and Biomedical Sciences, State University of New York, University at Buffalo, Buffalo, NY 14203, USA; 8Lion’s Eye Institute for Transplant & Research, Tampa, FL 33605, USA; david.ammar@lwvi.org; 9Department of Ophthalmology, Flaum Eye Institute, University of Rochester, Rochester, NY 14642, USA; karen_allison@urmc.rochester.edu; 10Department of Foundational Medical Studies and Eye Research Center, Oakland University William Beaumont School of Medicine, Rochester, MI 48309, USA; amtawfik@oakland.edu; 11Eye Research Institute, Oakland University, Rochester, MI 48309, USA; 12Department of Medicine (Biomedical Genetics), Boston University Chobanian & Avedisian School of Medicine, Boston, MA 02118, USA; sherva@bu.edu (R.M.S.); farrer@bu.edu (L.A.F.); 13Department of Biostatistics, Epidemiology and Informatics, Perelman School of Medicine, University of Pennsylvania, Philadelphia, PA 19104, USA; mingyao@pennmedicine.upenn.edu; 14Department of Ophthalmology, Perelman School of Medicine, University of Pennsylvania, Philadelphia, PA 19104, USA; stamboli@pennmedicine.upenn.edu; 15Retina Service, Massachusetts Eye & Ear, Department of Ophthalmology, Harvard Medical School, Boston, MA 02114, USA; ivana_kim@meei.harvard.edu; 16Genetics, Genomics and Bioinformatics Graduate Program, Jacobs School of Medicine and Biomedical Sciences, State University of New York, University at Buffalo, Buffalo, NY 14203, USA

**Keywords:** age-related macular degeneration (AMD), intermediate AMD (iAMD), neovascular AMD (NEO), macular retinal pigment epithelium (RPE)/choroid, macular retina, tissue-specific gene expression, splicing, allele-specific expression (ASE), microRNAs (miRNAs), and AMD therapies

## Abstract

Age-related macular degeneration (AMD) is a leading cause of blindness, and elucidating its underlying disease mechanisms is vital to the development of appropriate therapeutics. We identified differentially expressed genes (DEGs) and differentially spliced genes (DSGs) across the clinical stages of AMD in disease-affected tissue, the macular retina pigment epithelium (RPE)/choroid and the macular neural retina within the same eye. We utilized 27 deeply phenotyped donor eyes (recovered within a 6 h postmortem interval time) from Caucasian donors (60–94 years) using a standardized published protocol. Significant findings were then validated in an independent set of well-characterized donor eyes (*n* = 85). There was limited overlap between DEGs and DSGs, suggesting distinct mechanisms at play in AMD pathophysiology. A greater number of previously reported AMD loci overlapped with DSGs compared to DEGs between disease states, and no DEG overlap with previously reported loci was found in the macular retina between disease states. Additionally, we explored allele-specific expression (ASE) in coding regions of previously reported AMD risk loci, uncovering a significant imbalance in *C3* rs2230199 and *CFH* rs1061170 in the macular RPE/choroid for normal eyes and intermediate AMD (iAMD), and for *CFH* rs1061147 in the macular RPE/choroid for normal eyes and iAMD, and separately neovascular AMD (NEO). Only significant DEGs/DSGs from the macular RPE/choroid were found to overlap between disease states. *STAT1*, validated between the iAMD vs. normal comparison, and *AGTPBP1*, *BBS5*, *CERKL*, *FGFBP2*, *KIFC3*, *RORα*, and *ZNF292*, validated between the NEO vs. normal comparison, revealed an intricate regulatory network with transcription factors and miRNAs identifying potential upstream and downstream regulators. Findings regarding the complement genes *C3* and *CFH* suggest that coding variants at these loci may influence AMD development via an imbalance of gene expression in a tissue-specific manner. Our study provides crucial insights into the multifaceted genomic underpinnings of AMD (i.e., tissue-specific gene expression changes, potential splice variation, and allelic imbalance), which may open new avenues for AMD diagnostics and therapies specific to iAMD and NEO.

## 1. Introduction

Age-related macular degeneration (AMD) is a complex neurodegenerative disease with both intermediate and advanced forms and is the leading cause of visual disability in the aging population. The intermediate form may be a clinical biomarker indicating an increased risk of progression to either of the two advanced forms: neovascular AMD (also referred to as wet AMD), which correlates to rapid vision loss, and geographic atrophy (also referred to as dry AMD), in which it can take longer for vision loss to occur [1,2]. In either form, this condition involves the progressive degradation of the macula, leading to central vision loss, which impairs reading, facial recognition, and driving abilities [1].

While there is no cure for AMD, and treatments are limited in their efficacy, the development of anti-vascular endothelial growth factor (VEGF) has helped to mitigate visual loss associated with neovascular AMD, though it cannot fully restore anatomic or visual integrity [3,4,5]. While there are no FDA-approved therapies for intermediate AMD (iAMD), an over-the-counter supplementation of antioxidant AREDS2 formula has been demonstrated to modestly reduce the rate of progression to advanced AMD [6]. Therefore, our focus in the current study was intermediate AMD (iAMD) and neovascular AMD (NEO) to uncover pathways and mechanisms specific to these AMD subtypes. It is only through understanding disease mechanisms that appropriate therapies can be developed.

A large AMD genome-wide association study (GWAS) identified significant association at 34 loci [7] and directed us to potential pathways underlying disease mechanism(s). Studies that utilize single-cell, single-nuclei and/or bulk RNAseq are agnostic and unbiased approaches to examine gene expression, further adding value by demonstrating whether disease-associated loci are expressed in disease-affected tissue [8,9,10,11,12,13,14,15,16,17,18]. Additionally, RNA-Seq can uncover differentially spliced genes and non-coding RNAs [19,20,21,22,23,24,25]. The relationship between differential splicing and gene expression is crucial in shaping the proteome diversity observed in cells, with up to 95% of human multi-exon genes estimated to undergo alternative splicing [26]. Splicing fine-tunes gene expression via the generation of multiple protein isoforms from a single gene, yet our understanding of how splicing contributes to transcriptome variation is limited. Recent studies have explored how splicing diversity and gene expression vary across human traits, implicating aberrant splicing patterns in disease and illustrating the therapeutic potential of spliceosome-targeted therapies [27,28]. Thus, these findings demonstrate the contribution of differential splicing to differential gene expression and emphasize the need to investigate the interplay between these distinct yet interconnected mechanisms.

To date, RNAseq studies in AMD have not evaluated differential gene and splice expression simultaneously in the macular RPE/choroid and macular neural retina within the same well-characterized donor eye across disease states. We focused the present study on tissues specifically affected by AMD, the macula of the retinal pigment epithelium/choroid (RPE)/choroid and the macula of the neural retina, between iAMD, NEO and, separately, normal condition. To address the complexity of a multi-faceted disease like AMD, we utilized a systems biology approach, as previously employed [13,29,30,31]. In addition, motivated by previous studies showing evidence of allele-specific expression (ASE) [32] in genes associated with a risk of autism, stroke progression, Alzheimer disease and cancer [33,34,35,36,37,38], we interrogated the DNA of each donor for previously reported AMD GWAS coding variants [7]. This was undertaken to determine whether an imbalance of expression between alleles may underlie phenotypic variation, and hence the pathophysiology of AMD. To our knowledge, this is the first study to assess ASE across the clinical spectrum of AMD at a genome-wide level.

## 2. Resource Availability

### 2.1. Lead Contact

Requests for more information on the bulk data in the manuscript should be directed to Margaret M. DeAngelis (mmdeange@buffalo.edu).

### 2.2. Materials Availability

This study did not generate new unique reagents.

### 2.3. Data Availability

#### 2.3.1. Processed Data

Requests for more information on the RNAseq data in the manuscript should be directed to Margaret M. DeAngelis (mmdeange@buffalo.edu). The raw data reported in this study cannot be deposited in a public repository because of patient privacy reasons. De-identified human/patient details are listed in Table 1.

#### 2.3.2. Donor Eye Tissue Repository

Methods for human donor eye collection have been previously described in detail according to a standardized protocol [39], and moreover have been successfully used in several downstream genomics studies [13,40,41,42]. In brief, donor eyes were procured within a 6 h post-mortem interval time. Both eyes from each donor underwent post-mortem phenotyping with ocular imaging, including spectral domain optical coherence tomography (SD-OCT) and color fundus photography, as published. Retinal pigment epithelium/choroid was immediately dissected from the overlying retina, and the macula separated from the periphery using an 8 mm macular punch. For both peripheral and macular tissues, RPE/choroid was separated from the overlying retinal tissue using microdissection; tissue planes were optimized to minimize retinal contamination of RPE/choroid samples, and a subsequent 6 mm RPE/choroid tissue punch was taken from the 8 mm punch. For this experiment, retina and/or RPE/choroid tissues were placed in RNAlater (Ambion), an RNA stabilizing reagent stored as previously described [13]

In brief, AMD phenotyping employed the modified Age-Related Eye Disease Study severity grading scale, where AREDS category 0/1 was considered normal, AREDS category 3 was intermediate AMD (iAMD), and AREDS category 4b was neovascular AMD [43]. Phenotype analysis was performed as described [39], by a team of four retinal specialists and ophthalmologists at the University of Utah School of Medicine, Moran Eye Center, and the Massachusetts Eye and Ear Infirmary Retina Service. The agreement of all four specialists upon independent review of the color fundus and SD-OCT imaging was deemed diagnostic; discrepancies were resolved by a collaboration between a minimum of three specialists to ensure a robust and rigorous phenotypic analysis. Only one eye per donor was analyzed for further study. In the case of discordant phenotypes within the same donor, the more severe diseased eye was used for inclusion in the study. For example, if a patient had a diagnosis of AREDS 3 (iAMD) in one eye and AREDS 0/1 in the contralateral eye, only the AREDS 3 (iAMD) eye was used in the study. Similarly, if a patient had a diagnosis of AREDS 3 in one eye and neovascular AMD in the contralateral eye, only the neovascular eye was used in the study. In the case where donor eyes were concordant for disease status, one randomly selected eye per donor was chosen. Although AREDS category 2 (early AMD), category 4a (geographic atrophy) and AREDS category 4c (both geographic atrophy and neovascular AMD) were collected, they were not included in this study. Resultant transcriptomic and epigenomic data from these well-characterized donor eye phenotypes have been previously published [13].

### 2.4. Nucleic Acid Extraction and RNA-Sequencing

DNA and RNA were extracted from macular retina and macular RPE/choroid tissues using the Qiagen All-prep DNA/RNA mini kit (cat #80204) according to the manufacturer’s protocol from a total of 27 donors; 12 AREDS 0/1, 10 AREDS 3; 5 4b (neovascular) (a total of 54 samples). The quality of RNA samples was assessed with an Agilent Bioanalyzer. Total RNA samples were poly-A selected and cDNA libraries were constructed using the Illumina TruSeq Stranded mRNA Sample Preparation Kit (cat# RS-122-2101, RS-122-2102) according to the manufacturer’s protocol. Sequencing libraries (18 pM) were chemically denatured and applied to an Illumina TruSeq v3 single read flow cell using an Illumina cBot. Hybridized molecules were clonally amplified and annealed to sequencing primers with reagents from an Illumina pTruSeq SR Cluster Kit v3-cBot-HS (GD-401-3001). Following transfer of the flowcell to an Illumina HiSeq instrument (HCS v2.0.12 and RTA v1.17.21.3), a multiplexed, 50 cycle single read sequence run was performed using TruSeq SBS v3 sequencing reagents (FC-401-3002).

### 2.5. Primary Processing of RNA Sequencing Data

Each of the 54 samples (50 bp, poly-A selected, non-stranded, Illumina HiSeq) from the fastq datasets were processed as follows: reads were aligned using NovoCraft’s novoalign 2.08.03 software (http://www.novocraft.com/), accessed 15 July 2017, with default settings plus the -o SAM -r All 50 options to output multiple repeat matches. The genome index used contained human hg19 chromosomes, phiX (an internal control), and all known and theoretical splice junctions based on Ensembl transcript annotations. Additional details for this aspect of the protocol are described elsewhere (http://useq.sourceforge.net/usageRNA-Seq.html), accessed 15 July 2017.

Subsequently, raw novoalignments were processed using the open source USeq SamTranscriptiomeParser (http://useq.sourceforge.net), accessed 15 July 2017, to remove alignments with an alignment score greater than 90 (~3 mismatches), convert splice junction coordinates to genomic, and randomly select one alignment to represent reads that map equally well to multiple locations. Relative read coverage tracks were generated using the USeq Sam2USeq utility (http://useq.sourceforge.net/cmdLnMenus.html#Sam2USeq), accessed 15 July 2017, for each sample and sample type (Normal Retina, Intermediate AMD Retina, Neovascular AMD Retina, Normal RPE/choroid, Intermediate AMD RPE/choroid, and Neovascular AMD RPE/choroid).

Estimates of sample quality were determined by running the Picard CollectRNA-SeqMetrics application (http://broadinstitute.github.io/picard/), accessed 15 July 2017, on each sample. These QC metrics were then merged into one spreadsheet to identify potential outliers. Agilent Bioanalyzer RNA integrity number (RIN) and library input concentration columns were similarly added for QC purposes (http://www.genomics.agilent.com), accessed 15 July 2017.

### 2.6. Differential Gene Expression of Poly A Tail Sequencing and Splicing Analysis of Poly A Tail

Sample sets were analyzed using the DefinedRegionDifferentialSeq (DRDS) utility of USeq to detect differentially expressed and differentially spliced genes. This application accepts as input a conditions directory containing folders with biological replicates for each macular sample type (Normal Retina, Intermediate AMD Retina, Neovascular AMD Retina, Normal RPE/choroid, Intermediate AMD RPE/choroid, and Neovascular AMD RPE/choroid) and an Ensemble gene table in UCSC refFlat format. Gene models were created by merging gene transcripts into a single composite “gene” with the USeq MergeUCSCGeneTable utility. A table containing alignment counts from each sample for each gene was created with DRDS. Data in this table provided the basis for estimating count-based differential abundance using the DESeq2 Bioconductor package (http://www.bioconductor.org/packages/release/bioc/html/DESeq2.html), accessed 15 July 2017 and 23 August 2023 [44]. This program estimates the over-dispersion in the count data and calculates adjusted *p*-values using a negative binomial test. Benjamini–Hochberg *p*-value correction was applied to our adjusted *p*-values to control for multiple testing. Fold changes for differences in gene expression indicated the degree of change between conditions. DESeq2 also generates a log2 ratio estimate of difference in gene abundance using variance corrected counts as well as rLog values for clustering and principal component analysis (PCA). Library size and within-replica variance were estimated for each sample. Pairwise comparisons were made between the normal and disease subgroups. Differences in splicing were assessed for merged replica counts for each exon with ≥10 counts in each gene in each subgroup using a chi-square test. A Bonferroni multiple testing correction was applied and the exon with the biggest absolute log2 normalized gene count ratio was noted. Similarly, an adjusted *p*-value was calculated and fold changes for differences in splicing indicated the degree of potential splicing difference between conditions. A per-base normalized gene count read coverage log2 ratio graph was created, enabling the visualization of the relative exon coverage difference for each pairwise comparison. To identify potential outlier samples, unsupervised hierarchical clustering (HC) and PCA were performed with the aid of the Partek Genomic Suite (http://www.partek.com/pgs), accessed 15 July 2017 and 23 August, 2023, using the default settings. DESeq2 rLog values from genes with ≥20 counts were included in this pipeline. For HC visualization, row values were mean-centered at zero and scaled to a standard deviation of one. In addition to further demonstrating the quality of our data, violin plots of log_10_-transformed FPKM values were generated with the ggplot2 package (https://ggplot2.tidyverse.org/), accessed 11 September 2023 [45]. To display differential gene and splice expression, volcano plots were produced using freely accessible software (https://huygens.science.uva.nl/VolcaNoseR), accessed 19 May 2023 [46].

We used our previously published bulk RNAseq dataset from 85 donor eyes as our validation dataset [13]. Differential gene expression was performed as described above for DESeq2 and using the limma/voom package in R, as previously described [13,47]. Briefly, for the limma/voom package in R, normalization was carried out using TMM, controlling for age and sex in the analysis, and *p*-values were adjusted with Benjamini–Hochberg correction [47,48,49]. Genes were considered significant if they had an adjusted *p*-value less than 0.05 and a fold change ≥ 1.5 in either direction.

### 2.7. Bioinformatic Analysis

Gene Set Enrichment Analysis (GSEA v.4.3.2) software was utilized to profile our expression dataset (UC San Diego and Broad Institute, https://www.gsea-msigdb.org/gsea/index.jsp), accessed 6 September 2023 [50,51]. A gct file was created with our normalized FPKM count data, along with a phenotype cls file for input. The gmt file “h.all.v2023.1.Hs.symbols” and the chip file “Human_Ensembl_Gene_ID_MSigDB.v2023.1.Hs” were used in parallel with our input. The resulting output represented hallmarks or gene sets found to be enriched in our dataset. Based on the direction of our comparison (i.e., iAMD vs. Normal, NEO vs. Normal, etc.), genes that were higher-ranking or more associated with the phenotype on the left contributed positively to the enrichment score (ES) and the lower-ranking genes or genes associated to a lesser extent with the phenotype contributed negatively to the ES. This enrichment score was then normalized based on the variation in our gene set, as it was by others [51], giving us our normalized enrichment score (NES).

Next, we examined genes previously demonstrated to be associated with AMD in candidate gene studies and/or GWAS conducted by the International AMD Genetics Consortium (IAMDGC) and Gorman et al. (2022) [7,29,30,52,53,54,55,56,57,58,59] for overlap with our DEGs or DSGs. Genes that overlapped (e.g., were both differentially spliced and differentially expressed) were then validated for expression between the same tissue type and disease comparison in a different macular bulk RNAseq dataset from our lab [13]. We further explored how the validated genes we identified may be regulated via the UCSC genome browser [60].

QIAGEN Ingenuity Pathway Analysis (IPA) (QIAGEN Inc., Redwood City, CA 94063, United States, (https://digitalinsights.qiagen.com/IPA), software accessed on 8 October 2023, was utilized, as previously described, for the functional analysis of our validated genes (significant differential expression, significant differential splice expression, and significant in a second RNAseq dataset) [30,61]. An IPA-generated network was created to visualize interconnectedness between our validated genes. This network was overlaid with our differential expression data and differential splice data separately.

All bar chart representations were created using GraphPad Prism, version 10.0.2 for macOS (GraphPad Software, Boston, MA, USA, www.graphpad.com). All Venn diagrams were generated using InteractiVenn (http://www.interactivenn.net/index2.html), accessed 12 September 2023 [62].

### 2.8. Allele-Specific Expression (ASE)

SNPs previously identified by GWAS as being associated with AMD (determined using the GWAS Catalog, accessed 15 July 2017, https://www.ebi.ac.uk/gwas/) [7] were investigated for allele-specific expression (ASE) in our dataset. Specifically, we genotyped the exonic AMD SNPs using either the genotypes from the HumanOmni2.5-8 BeadChip Kit or TaqMan assays. Bam files of individuals showing a heterozygous genotype were examined to determine the number of reads for each of the two alleles. Genotypes of heterozygotes determined from the SNP Chip showing monoallelic expression were confirmed using proxies (r^2^ ≥ 0.8), as determined by the 1000 Genomes phase 3 CEU reference panel. Only individuals with ≥10 reads were used. A binomial test, corrected using Benjamini–Hochberg, was used to determine statistically significant allelic imbalance within each individual [63].

### 2.9. Differential Expression Validation with Real-Time PCR

RNA was reverse-transcribed using oligo-dT primers (Invitrogen, 5781 Van Allen Way Carlsbad, CA 92008, United States) and SuperScript III reverse transcriptase (Invitrogen, 5781 Van Allen Way Carlsbad, CA 92008, United States), according to the manufacturer’s protocol. Then, cDNA was used as a template for real-time PCR reactions, and run in triplicate using pre-designed Taqman Gene Expression Assays (Life Technologies, 5781 Van Allen Way Carlsbad, CA 92008, United States) for *UCHL1*, *PFKP*, *LPCAT1*, *PDPN*, *GAS1*, and *CST3*, and for *UBC* as an endogenous control. Assays were run on the Taqman 7500 Real Time PCR system (Life technologies). Mean Ct values were normalized to *UBC* and analyzed using the 2^-∆∆CT^ method, as previously described [64].

## 3. Results

Numbers and characteristics of subjects available for the analysis of each tissue type after QC are shown in Table 1. For clustering analysis, using both hierarchal clustering (HCA) and principal component analysis (PCA) based on the samples’ whole transcriptome expression, samples were split into two primary groups, comprising retina and RPE/choroid samples. A clear separation of macular RPE/choroid and macular retina tissue types was observed. Hierarchal clustering demonstrated greater variability among the macular RPE/choroid samples than for the macular retina samples. Log_10_-transformed FPKM values were plotted and demonstrated no significant difference in our overall count data between groups (Figure 1). Out of the 54 samples, 47 samples passed QC analysis (Table 1).

To evaluate the quality of our tissue dissection, we calculated the number of reads mapped to genes known to be expressed exclusively in the neural retina and RPE/choroid, respectively, using an approach as previously described for the retina [65]. Retina genes involved in phototransduction (*GNGT1*, *GUCA1A*, *PDE6A*, *GNB1*, *CNGB1*, *GNAT1*, *CNGA1*, *PDE6B*, *PDE6G*, *PRPH2*, *RHO*, *ROM1*, *SAG*, and *SLC24A1*) accounted for an average of 2.3% of reads in the total only in the normal retina library, and accounted for only 0.06% of our normal RPE/choroid tissue reads, proportions which are similar to those reported in a previous study [66]. In our study, RPE/choroidal genes (*BEST1*, *RDH5*, and *RPE65*) accounted for an average of 0.65% of reads in the total RPE/choroid library and only 0.02% of total reads in the retina library. These findings demonstrate that neither the macular retina nor the macular RPE/choroid was relevantly contaminated (e.g., if there was contamination of the retina genes in the RPE/choroid library, reads would be greater than 1% compared to the observed proportion of 0.06%). In addition, we plotted our log_10_ -transformed FPKM values and showed a similar distribution across our sample conditions, illustrating that our expression results were not due to sample variability.

### 3.1. Gene Expression Differences

A total of 26,650 genes were expressed in the macula RPE/choroid and/or macula retina. Within phenotyped normal eyes, 16,638 genes showed significant (FDR ≤ 0.05) differential expression between macular RPE/choroid and macular retina tissues with a minimum fold change ≥ |1.5|. As illustrated in Figure 2, within macular RPE/choroid tissues, significant differential expression was observed for 40 genes between iAMD and normal eyes, 1204 genes between NEO and normal eyes, and 1194 genes between iAMD and NEO eyes (Figure 2A–C, Appendix A). Within macular retina tissues, 30 genes were differentially expressed between iAMD and normal eyes, 41 genes were differentially expressed between NEO and normal eyes, and 50 genes were differentially expressed between iAMD and NEO donor eyes (Figure 2D–F).

Of these differentially expressed genes in the macula RPE/choroid, 29 were unique to iAMD vs. normal, 285 were unique to NEO vs. normal, and 276 were unique to iAMD vs. NEO (Appendix A). Of the 40 significant DEGs in the iAMD vs. normal comparison of the macular RPE/choroid and the 1204 significant DEGs in the NEO vs. normal comparison of the macular RPE/choroid, only six genes (*MTRN2L1*, *CLEC2L*, *CCM2L*, *CYP4X1*, *GLDN*, and *SMAD7)* were found to overlap (Figure 3A). However, none of the above genes were found to be statistically significant in the iAMD vs. NEO comparison of the macular RPE/choroid.

Genes that were unique to the macular retina in iAMD vs. normal (*n* = 27), NEO vs. normal (*n* = 38), and iAMD vs. NEO had not been previously associated with AMD (Appendix A). Of the 30 significant DEGs in iAMD vs. normal of the macular retina and the 41 significant DEGs in neovascular AMD vs. normal of the macular retina, only two genes (*FRG1* and *CERKL)* were found to overlap (Figure 3B). However, none of the above genes were found to be statistically significant in iAMD vs. NEO in the macular retina.

Of note, only one gene, mitochondrial-derived peptide humanin *MTRNR2L1* [67], overlapped between any RPE/choroid and retina disease comparisons (Appendix A). *MTRNR2L1* was found to be differentially expressed in iAMD vs. normal for both the macular neural retina and macular RPE/choroid.

A total of nine unique microRNAs (miRNAs) were identified (*MIR146A*, *MIR3918*, *MIR4657*, *MIR17HG*, *MIR3620*, *MIR3064*, *MIR197*, *MIR4680*, and *MIR4647*) across all disease comparisons. Of these miRNAs, six (*MIR4657*, *MIR17HG*, *MIR3620*, *MIR197*, *MIR3064*, and *MIR3918*) were found to be differentially expressed in iAMD vs. NEO within the macular RPE/choroid, while three (*MIR146A*, *MIR197*, and *MIR3918*) were identified in NEO vs. normal in the macular RPE/choroid (Appendix A). In the macular retina, two miRNAs were observed: *MIR4680*, in neovascular AMD vs. normal, and *MIR4647* in intermediate AMD vs. NEO (Appendix A). Also noteworthily, a unique lncRNA (*AC000124.1*) was downregulated in iAMD compared to NEO RPE/choroid, while *PIWL1* was upregulated in NEO compared to normal macular RPE/choroid (Appendix A).

### 3.2. Gene Splicing Differences

Similar to the DEG results, the highest number of DSGs in our RNAseq data was observed in NEO vs. normal, with 1154 significant DSGs in the macular retina and 629 in the macular RPE/choroid (Figure 4B,E). Similar to the DEG analysis, fewer DSGs were observed for comparisons of iAMD vs. NEO (810 in the macular retina, 608 in macular RPE/choroid; Figure 4C,F). The lowest number of DSGs was identified in the iAMD vs. normal comparison (210 in the macular retina, 177 in the macular RPE/choroid; Figure 4A,D). When comparing DSGs between the macular retina and the macular RPE/choroid for each disease comparison, there were 13 DSGs that overlapped between tissue types in iAMD vs. normal, 152 DSGs overlapping between NEO vs. normal, and 102 DSGs overlapping between iAMD vs. NEO (Appendix A).

For the macula RPE/choroid, there were 26 significant DSGs overlapping all three disease state comparisons in the RPE/choroid and 56 DSGs overlapping between iAMD vs. normal and NEO vs. normal. Of these 56 genes, 8 genes, *CCPG1*, *GALNT15*, *PLEKHA*, *RGS20*, *TMEM14B*, *ULK4*, *VPS13C*, and *VPS37A*, were not regulated in the same direction, this resulted in 48 (identically regulated) overlapping genes (Figure 3C). When examining DSGs unique to each disease state comparison, there were 94 in intermediate AMD vs. normal macula RPE/choroid, 247 in iAMD vs. NEO in RPE/choroid, and 263 in NEO vs. normal in macula RPE/choroid (Appendix A). When evaluating DSGs in the macular retina, 68 genes overlapped between iAMD vs. normal and NEO vs. normal. Of these 68 genes, 9 genes, *CCT2*, *CNOT2*, *EXOC3*, *GPATCH2*, *HDAC9*, *KIAA1841*, *RPF2*, and *SNHG14*, were not regulated in the same direction, this resulted in 59 (identically regulated) overlapping genes (Figure 3D).There were 115 genes unique to iAMD vs. normal, 645 to NEO vs. normal comparison, and 329 unique to iAMD vs. NEO (Appendix A).

### 3.3. Gene Set Enrichment Analysis Using Our Normalized Expression Dataset

Gene Set Enrichment Analysis (GSEA) was employed to identify statistically significant hallmarks (*p*-value < 0.05, FDR q-value ≤ 0.05) enriched in our expression dataset found in both the macular RPE/choroid and macular retina (Figure 5). When utilizing normalized FPKM counts from the macular RPE/choroid of iAMD compared to normal, nine hallmarks (Figure 5A) were found to be upregulated in iAMD, with Wnt/β-catenin signaling at the top (Wnt/β-Catenin Signaling, Notch Signaling, Myogenesis, Hedgehog Signaling, Apical Junction, UV Response Dn, Kras Signaling Dn, TGF-β Signaling, Apical Surface). Continuing with the macular RPE/choroid, 10 hallmarks (Figure 5B) were found to be significantly upregulated in NEO compared to normal, with Angiogenesis having the highest normalized enrichment score (NES) (Angiogenesis, TGF-β Signaling, Notch Signaling, Unfolded Protein Response, Fatty Acid Metabolism, Wnt/β-Catenin Signaling, Apical Junction, Myogenesis, Adipogenesis, TNFα Signaling via NFKB). Interestingly, when comparing iAMD to NEO, three hallmarks (Figure 5C) were found to be significantly upregulated in iAMD (Spermatogenesis, Hedgehog Signaling, Pancreas Beta Cells), while seventeen hallmarks (Figure 5C) were found to be statistically significant in NEO (Interferon Gamma Response, TNFα Signaling Via Nfkb, Interferon Alpha Response, Oxidative Phosphorylation, Myc Targets V1, Unfolded Protein Response, Adipogenesis, Fatty Acid Metabolism, IL6 Jak/Stat3 Signaling, Reactive Oxygen Species Pathway, Myc Targets V2, P53 Pathway, Inflammatory Response, Uv Response Up, Xenobiotic Metabolism, Mtorc1 Signaling, Dna Repair).

Subsequently, we examined the macular retina with GSEA in an equal manner. Only one hallmark (Oxidative Phosphorylation) was identified to be significantly upregulated in iAMD vs. normal in the macular retina (Figure 5D). No hallmarks were found to be statistically significant in NEO compared to normal based on these parameters. When comparing iAMD to NEO, five hallmarks were found to be statistically significant (Figure 5E). Two hallmarks (Apical Surface, Pancreas Beta Cells) were upregulated in iAMD and three hallmarks (Hypoxia, Angiogenesis, IL6 Jak/Stat3 Signaling) were found to be significant in NEO (Figure 5E).

### 3.4. Analysis of DEGs and DSGs for Overlap with Genes Previously Associated with AMD

#### DEGs and DSGs: Normal Macular RPE/Choroid vs. Normal Macular Retina

Genes demonstrated to be associated with AMD risk in prior candidate or GWAS studies were examined [7,29,30,52,53,54,55,56,57,58,59] in our normal tissue, comparing the macular RPE/choroid to the macular retina, to characterize the transcriptomic landscape at a baseline state in the tissue affected by disease. A total of 94 DEGs out of 115 of the previously identified AMD loci were found to have statistically significant differential expression in our data set (Table 2). Sixty-seven DEGs (*ABCA1*, *ABHD2*, *ACAA2*, *ADAMTS9-AS1*, *ADAMTS9-AS2*, *C2*, *C3*, *C4A*, *C5*, *C9*, *CD46*, *CD55*, *CD63*, *CETP*, *CFB*, *CFH*, *CFHR3*, *CFI*, *CNN2*, *COL5A1*, *COL8A1*, *EXOC3L2*, *FILIP1L*, *FLT1*, *HLA-DQB1*, *IER3*, *IGFBP7*, *IL6*, *ITGA7*, *LBP*, *LIPC*, *LRP6*, *ME3*, *MMP19*, *MMP9*, *MYO1E*, *NPLOC4*, *OCA2*, *PCOLCE*, *PDGFB*, *PILRA*, *PKP2*, *PLA2G4A*, *RAD51B*, *RASIP1*, *RDH5*, *RGS13*, *RLBP1*, *ROBO1*, *RRAS*, *SERPINA1*, *SKIV2L*, *SLC16A8*, *SMAD3*, *STON1*, *STON1-GTF2A1L*, *TGFB1*, *TGFBR1*, *TIMP3*, *TNF*, *TNFRSF10A*, *TRPM1*, *TRPM3*, *TSPAN10*, *TYR*, *UNC93B1*, and *VDR*) were shown to have significantly higher expression (*padj* < 0.05) in the macular RPE/choroid (Table 2). Less than half, or 27 DEGs (*ABCA7*, *ADAM19*, *AFF1*, *ARHGAP21*, *B3GALTL*, *C10orf88*, *CCT3*, *CDH7*; *CDH9*; *CLUL1*; *CSK*; *CYP24A1*; *DDR1*; *HERC2*; *HTRA1*; *KMT2E*; *NLRP2*, *PELI3*, *RORα*; *RORβ*, *RP1L1*, *SPEF2*, *SRPK2*, *SYN3*, *TMEM97*, *VTN*, and *ZNF385B*), were observed with significantly higher expression in the normal macular retina compared to the normal macular RPE/choroid (Table 2). In contrast, about half (or twelve) DSGs (*ADAM19*, *CCT3*, *CD55*, *CLUL1*, *GTF2A1L*, *MMP9*, *PCOLCE*, *RORα*; *SPEF2*, *SRPK2*, *TGFB1*, and *ZBTB38*) were shown to have significantly higher expression (*padj* < 0.05) in the macular RPE/choroid (Table 3). Fourteen DSGs (*ABHD2*, *AFF1*, *ARHGAP21*, *C2*, *CD63*, *FILIP1L*, *FLT1*, *PILRA*, *RDH5*, *RLBP1*, *STON1-GTF2A1L*, *TRPM1*, *TRPM3*, and *TSPAN10*) were observed with significantly higher expression in the normal macular retina compared to the normal macular RPE/choroid (Table 3). Significant DEGs/DSGs in normal macular RPE/choroid vs. normal macular retina illustrated the direction of expression in each dataset (Figure 6).

### 3.5. DEGs: Macular RPE/Choroid Disease State Comparisons

Regarding previously identified AMD loci [7,29,30,52,53,54,55,56,57,58,59], *CDH7* was previously reported to have a suggestive association with AMD in non-smokers using GWAS [59]. We observed that *CDH7* was found to be significantly lower in expression in iAMD vs. normal RPE/choroid macular tissues [59]. Expression was significantly higher for *ABCA7* (*padj* = 0.002) and *RORα* (*padj* = 0.004) in NEO compared to normal in the macular RPE/choroid (*padj* ≤ 0.01; Appendix A) [30,54,55,56]. *VTN* expression (*padj* = 0.02) was also found to be significantly higher in NEO compared to normal in the macular RPE/choroid (Table 4) [7]. Interestingly, *FLT1* was the only previously associated AMD gene to be significantly lower in NEO compared to normal macula in the RPE/choroid (Table 4) [68].

Consistent with these results, we found that the expression of *ABCA7* (*padj* = 0.002) was significantly lower in iAMD macular RPE/choroid compared to NEO macular RPE/choroid (Table 4) [54]. Expression was also found to be significantly higher for *TNFRSF10B* (*padj* = 0.02) and *TRPM1* (*padj* = 0.04) in iAMD macular RPE/choroid compared to NEO macular RPE/choroid [7]. Additionally, *SPEF2* expression (*padj* = 0.02) was significantly lower in iAMD macular RPE/choroid compared to NEO macular RPE/choroid (Table 4) [7].

### 3.6. DEGs: Macular Retina Disease State Comparisons

As noted in Table 4, there was no overlap between significant DEGs observed between disease states within macula retina tissues for previously reported AMD loci [7,29,30,52,53,54,55,56,57,58,59].

### 3.7. DSGs: Macular RPE/Choroid Disease State Comparisons

Regarding the overlap of our DSGs with previously reported AMD loci [7,29,30,52,53,54,55,56,57,58,59], *CFB* and *FLT1* were up regulated while *C2* and *CLUL1* were downregulated in the macular RPE/choroid of iAMD vs. normal (*padj* < 0.05). For the DSG comparison of NEO vs. normal, *CFB* and *ABHD2* were upregulated in NEO, while *RORα*, *ABCA7*, *CLUL1* and *AFF1* were downregulated in this same disease comparison. For iAMD vs. NEO comparison in the macular RPE/choroid, *CFB*, *RORα*, *SPEF2*, *CLUL1* and *CDH9* were upregulated, whereas *ABHD2* was downregulated in this same disease comparison (Table 5).

### 3.8. DSGs: Macular Retina Disease State Comparisons

When comparing significant DSGs from iAMD vs. normal macular retina with those previously associated with AMD [7,29,30,52,53,54,55,56,57,58,59], *CCT3*, *CDH9*, and *ACAD10* were downregulated in iAMD (Table 6). *SPEF2*, *C3*, *CLUL1*, *ZNF385B*, *SMAD3*, and *ME3* were significantly upregulated in NEO vs. normal macula retina, while *ARHGAP21*, *ROBO1*, *TRPM1*, *LRP2*, and *HERC2* were significantly downregulated in NEO vs. normal in the macular retina (Table 6). Seven DSGs (*ARHGAP21*, *COL4A3*, *SKIV2L*, *TRPM1*, *LRP2*, *HERC2*, and *ADAM19*) were significantly upregulated in iAMD vs. NEO macular retina [7,53,56] (Table 6), while *SPEF2*, *ZNF385B*, and *SMAD3* were downregulated in iAMD vs. NEO (Table 6).

## 4. Overlap of Differentially Expressed Genes and Differentially Spliced Genes

Utilizing our systems biology approach to drill further down to the disease mechanism in AMD pathophysiology, we looked for overlap in our DEGs and DSGs in each disease comparison and tissue type. Only a small proportion of the overall DSGs in the macular RPE/choroid were also found to be DEGs in each disease state comparison: 6 significant genes in iAMD vs. normal macular RPE/choroid; 162 NEO vs. normal macular RPE/choroid; and 137 iAMD vs. NEO macular RPE/choroid. Of the DSGs in the macular RPE/choroid, 97 overlapped between NEO vs. normal and iAMD vs. NEO RPE/choroid comparisons (Appendix A). When comparing DSGs to the DEGs in the retina, there were no overlapping DSGs and DEGs.

## 5. Validation of Overlapping DEGs and DSGs through Bulk RNAseq

To validate our findings from the overlapping DEG/DSGs, we utilized our previously published bulk macula RNAseq dataset of well-characterized donor eye tissue [13]. As noted in the methods, we reanalyzed this data set so that the same comparisons could be made. We found no statistically significant differentially expressed genes when comparing iAMD vs. NEO in the macula RPE/choroid in our bulk RNAseq dataset [13]. Only one DEG/DSG, *STAT1*, was validated in our iAMD vs. normal macular RPE/choroid comparison (Table 7). Seven DEG/DSGs (*AGTPBP1*, *FGFBP2*, *CERKL*, *BBS5*, *RORα*, *ZNF292*, and *KIFC3*) were validated in the bulk RNAseq data set in NEO vs. normal macula RPE/choroid. Of these, only gene variants in *RORα* have been previously associated with AMD risk [7,30,55,56] (Table 7). Additionally, of these validated genes, the proteins of *AGTPBP1*, *CERKL*, *BBS5*, and *KIFC3* were expressed in the cytoplasm as opposed to the nucleus. Using the UCSC Genome Browser, we found that all of our validated DEG/DSGs contained numerous transcription-factor binding sites at the splice site coordinates identified. In addition, DEG/DSG *ZNF292* splice coordinates overlapped with a transcription start site (TSS).

When interrogating the seven validated DEG/DSGs (*AGTPBP1*, *BBS5*, *CERKL*, *FGFBP2*, *KIFC3*, *RORα*, and *ZNF292*) from the NEO vs. normal comparison in Ingenuity Pathway Analysis (IPA), a network was generated of upstream regulators and downstream targets forming possible interconnections between these DEG/DSGs (Figure 7A,B). Eleven miRNAs were found to form relationships with these genes. Next, *STAT1*, from our iAMD vs. normal comparison, was added to the network to visualize how it may interact with our seven validated genes (Figure 7C,D). All networks were overlaid with our expression data corresponding to the respective disease state (iAMD or NEO) for DEGs and DSGs (Figure 7A–D).

## 6. Allele-Specific Expression (ASE) of Known AMD-Associated SNPs

According to annotation information for published AMD genome-wide association studies included in the NHGRI-EBI Catalog (https://www.ebi.ac.uk/gwas/home; accessed 28 March 2017), 12 AMD-associated SNPs are located in coding regions of *APOE*, *ARMS2* (1), *C2* (1), *C3* (2), *CFB* (1), *CFH* (4), *CFI* (1), and *PLA2G12A* (1), and therefore were investigated for allele-specific expression (Table 8). No heterozygotes were found in our samples for *CFH* rs121913059, *CFI* rs141853578, *C3* rs147859257, or *APOE* rs429358. We found no expression of *ARMS2* in either the RPE/choroid or neural retina, and therefore we could not investigate the coding SNP rs10490924. For those heterozygotes showing mono-allelic expression (*n* = 6), *CFH* rs10754199 was used to confirm heterozygotes for *CFH* coding SNPs rs10661170 and rs1061147 (r^2^ = 1 for rs10754199 and both coding SNPs), *CFB* rs2242572 was used to confirm heterozygous genotypes for *CFB* rs641153 (r^2^ = 1), and *C3* rs1047286 was used to confirm the heterozygous genotype of *C3* rs2230199 (r^2^ = 0.843). Significant ASE was detected within individuals in four SNPs: *CFH* rs1061170 (Y402H), *CFH* rs1061147, *CFB* rs641153, and *C3* rs2230199. Specifically, for *CFH* rs1061170 we found significant ASE within 2/6 intermediate AMD RPE/choroid samples, and within 1/7 normal RPE/choroid samples. None of the four neovascular AMD RPE/choroid heterozygotes showed ASE, indicating that there were 10 or fewer reads for these samples in the retina data. For *CFH* rs1061147, significant ASE was observed within 5/6 intermediate AMD RPE/choroid samples, 3/4 neovascular RPE/choroid samples, and 7/7 normal RPE/choroid samples. These same heterozygotes had 10 or fewer reads among the retina data. The single heterozygote for *CFB* rs641153 (a normal sample) showed significant ASE within the RPE/choroid tissue. There were 10 or fewer reads for this SNP in the macula retina. There was significant ASE for *C3* rs2230199 within 2/3 intermediate AMD RPE/choroid samples, 0/1 neovascular AMD RPE/choroid samples, and 2/2 normal RPE/choroid samples. These same heterozygotes had 10 or fewer reads in the retina tissue.

## 7. Validation and Replication of RNAseq Findings

We validated our RNA-Seq methodology by choosing genes that varied in fold expression from a range of +20 to −20 (FDR of *p* < 0.05) between the normal RPE/choroid and retina—*UCHL1*, *PFKP*, and *LPCAT1* (down-regulated in RPE/choroid vs. retina) and *PDPN*, *GAS1*, and *CST3* (up-regulated in RPE/choroid vs. retina)—using real-time qPCR reactions run in triplicate on a subset of samples that were used for the RNAseq experiments. We confirmed the direction of effect for five of the six genes examined (Appendix A). We were unable to detect *PFKP* expression in all of the RPE/choroid tissue, and therefore this gene could not be validated. Additionally, we were able to replicate all our top 20 genes from the normal RPE/choroid vs. normal retina with the Human Eye Integration data (https://eyeintegration.nei.nih.gov/), accessed 15 July 2017. This database is a collection of healthy human RNAseq datasets generated from various studies of human eye tissue. To the best of our knowledge, no public database is yet available that contains gene expression data of macular retina and macular RPE/choroid tissues from the same donor eyes across the different clinical stages of AMD.

## 8. Discussion

In this study, we utilized a global RNAseq approach to investigate gene, splice, and allele-specific expression profiles in the macular retinal pigment epithelium/choroid (RPE/choroid) and the macular retina of post-mortem eyes from individuals with intermediate AMD or neovascular AMD, comparing them to normal age-matched controls. Additionally, this is the first study of its kind to compare macular RPE/choroid and macular retina in this manner within the same deeply phenotyped donor eye (obtained in a post mortem interval time < 6 h).

While it is clear that both the macular RPE/choroid and macular retina are important to AMD pathophysiology, studies have hypothesized that macular RPE/choroid cell function is more significantly related to AMD pathophysiology than retinal cell function [7,13,65,69,70,71,72]. However, there are only a few studies that compare the pathological changes occurring in the macula area (RPE/choroid) to the inner retina [69,73], even though AMD predominantly impacts the macula; thus, the lack of information leaves the relationship between the tissue types at given stages of disease unclear. In this study, at a global RNAseq level, we show the importance of gene expression, splicing, and allele-specific expression in the macular RPE/choroid compared to the macular retina in iAMD and NEO pathophysiology. The bulk RNASeq approach allowed us to identify genes that otherwise would not have been identified via a single-nuclei RNASeq approach, as their expressed transcripts are located in various cellular compartments, including the cytoplasm for validated genes: *AGTPBP1*, *CERKL*, *BBS5*, and *KIFC3* (https://www.genecards.org/), accessed 28 August 2023. However, this was also a limitation of our study, as a single-cell approach may have been more appropriate due to the diverse cell types that comprise the macular retina. Therefore, we may have failed to identify significant gene expression changes critical to AMD pathophysiology in the macular retina, as described as by others [15,16,74,75].

As previously demonstrated by others, across human tissues [28] and in neurodegenerative conditions [76,77], there is very little overlap between DEGs and DSGs; thus, these two sets of biological processes appear to operate through distinct mechanisms. This could be due to the fact that the majority of splice isoforms undergo nonsense-mediated decay [78] and do not become functional proteins, which may be the case for some of the differentially spliced genes identified herein. Additionally, techniques used to detect differential splicing compared to differential gene expression have different sets of biases and therefore are inherently noisier compared to overall gene expression changes [79]. Our data suggest that AMD’s genomic underpinnings are multifaceted and may involve various regulatory mechanisms that require further exploration.

In this study, we only found overlap between DEGs and DSGs in the macular RPE/choroid between any disease state comparison (Appendix A). Furthermore, we validated a handful of our DEG/DSGs in an independent bulk RNAseq data set. *STAT1* was significantly increased in iAMD compared to normal in our DEGs and an independent bulk RNAseq data set [13], but was significantly decreased in our DSGs (Table 7). Notably, *STAT1* was the only validated gene in our iAMD vs. normal comparison, and we hypothesize the difference between its DEG and DSG state may be a compensatory mechanism during the intermediate stage of disease development. Studies have also tied interferon-*γ* to *STAT1* signaling, where interferon-*γ* has been connected to RPE cell death [80] and shown to negatively regulate *HTRA1* expression by activating the p38 MAPK/STAT1 pathway; further it has been shown that *STAT1* can bind the *HTRA1* promoter [81]. Additionally, interferon-*γ* signaling was also an enriched hallmark in our GSEA analysis. In NEO vs. normal, we validated seven genes (*AGTPBP1*, *BBS5*, *CERKL*, *FGFBP2*, *KIFC3*, *RORα*, and *ZNF292*) from our DEGs, DSGs, and in an independent bulk-RNAseq dataset (Table 7). Independently, these genes have been linked to AMD (*RORα* [30,55]), eye disease (*BBS5* [82], *CERKL* [83,84,85], *KIFC3* [86,87]) and other immune/neurodegenerative conditions (*AGTPBP1* [88], *FGFBP2* [89,90], *KIFC3* [91,92], *ZNF292* [93]). As previously reported, *RORα* has also been shown to interact with the *AMRS2/HTRA1* locus [30,55]. In addition, *BBS* genes [94], *RORα* [95], and the circular RNA of *ZNF292* [96] have been demonstrated to interact with the Wnt/β-Catenin signaling pathway, which we found as an enriched hallmark (Figure 5) in our AMD disease states (iAMD and NEO). We previously found *RORα* to be downregulated in peripheral blood samples from patients with NEO [30], but when investigating the affected tissue herein, we found it to be upregulated in our DEGs and further validated in our independent bulk RNAseq dataset [14] (Table 7). This further highlights the need for additional studies exploring the relationship between biomarkers identified in patient samples of serum/blood and the actions of those markers in disease-affected tissue, as we still do not know whether AMD is a local or systemic disease [97,98]. As illustrated in our network, an interesting picture emerged when we investigated our seven validated genes in the NEO vs. normal comparison (Figure 7A,B). The bulk of the interconnections between these seven genes were miRNAs, predicted to have an inhibitory effect. Once we added STAT1 (from iAMD vs. normal), this continued to remain the case (Figure 7C,D). Thus, we hypothesize that miRNA degradation may be underlying disease development of AMD and remains an avenue to be explored.

Various factors in addition to gene and splice expression, such as microRNAs (miRNAs), RNA-binding proteins, and long non-coding RNAs (lncRNAs), may influence transcript stability and modulate translation at a tissue-specific level [98]. As discussed above, once we generated a network(s) for our cross validated genes (DEGs, DSGs, and an independent bulk RNAseq dataset), relationships were primarily seen with miRNAs. Thus, miRNAs may represent a potential therapeutic target for diagnosis, prognosis and/or treatment [98,99]. *MiRNA-146a* was downregulated in macular RPE/choroid donor tissues from neovascular AMD subjects compared to controls (Appendix A). Previously, *miRNA-146a* was shown to be upregulated in the serum of patients with neovascular AMD [100,101,102], again underscoring the importance of tissue specificity with gene expression. Targets of *miRNA-146a* have been implicated in the modulation of the immune response in endothelial tissue, including in the negative regulation of complement factor H [103,104,105]. While the involvement of the non-coding genome is under active investigation, miRNAs and other non-coding RNAs (lncRNA-*AC000124.1* and *PIWL1* in our results-Appendix A) have been found to have key roles in cellular homeostasis, with disruption leading to human diseases such as cancer [106]. Further studies need to be conducted to fully characterize the role of lncRNAs and miRNAs as biomarkers and determine their potential as therapeutic targets. In this study, we demonstrated that genes previously associated with AMD risk, relevant signaling pathways, and miRNAs and other ncRNAs are expressed differently between tissue types and disease states.

*MTRNR2L1* was the only gene found to overlap between the macular retina and macular RPE/choroid in any disease state comparison (found in iAMD vs. normal). While it is known that *MTRNR2L1* is a nuclear-encoded humanin isoform gene, its biological function is currently unknown. Recent studies have demonstrated that humanin, a small peptide derived from the mitochondria, can protect the RPE cells against mitochondrial damage induced by oxidative stress and endoplasmic reticulum stress [107,108]. The upregulation of *MTRNR2L1* in the retina and the RPE/choroid tissues with AMD indicates a potential role of *MTRNR2L1* in protecting against retinal and RPE damage during disease development and the progression of AMD. However, the exact role of this gene warrants future investigation.

While we highlight above the need to explore the relationship between genome-wide association studies (GWAS)’s loci and gene expression, determining the causative genes responsible is another matter altogether [13]. The challenge arises from a few factors. First, a significant proportion of these SNP associations are found in regions of the genome that are non-coding, known as intra- or intergenic regions [109]. Second, each identified association may involve more than one candidate gene [110]. Thirdly, gene expression is highly tissue- and cell-specific [111], so what is found in these GWAS studies is not necessarily found in the macular RPE/choroid or macular retina. Thus, when considering previously identified AMD risk loci [1,7,29,68,112], it was interesting that 67 of our DEGs (Table 2) showed significantly higher expression in the normal macular RPE/choroid compared to the normal macular retina, underscoring the importance of tissue and geographic location [10,13,15,71]. For the DSGs, only a small portion overlapped with previously reported AMD loci, and these were fairly evenly distributed between the normal macular RPE/choroid and normal macular retina in expression differences (Table 3). DEGs that were found to have a higher expression in the normal macular RPE/choroid were found to have a higher expression as DSGs in the normal macular retina (Figure 6). This observed bidirectional change in genes that overlapped in the macula suggests an expression-dependent, homeostatic mechanism in unaffected tissue. We did not find a large portion of the previously identified AMD risk loci differentially expressed or spliced in our disease states of iAMD and NEO when compared to each other, or separately to normal (Table 4, Table 5 and Table 6). No previously associated AMD genes were found to overlap with our DEGs in the macular retina when considering disease state, but previously associated AMD genes were found to overlap with our DSGs in the macular retina at a higher number compared to the macular RPE/choroid (Table 6). This could be due to a lack of tissue specificity [28] for differential splicing in the macula, which is further supported by our normal macular RPE/choroid vs. normal macular retina DSG findings (where an approximately equivalent expression between macular RPE/choroid and macular retina was observed). Of note, while a prior report demonstrated that a splice variant in *TRPM1* was expressed more highly in the retina of late-stage AMD donor eyes and a second splice variant in *TRPM1* was expressed in the RPE/choroid of AMD donor eyes [14], we did not find splicing for *TRPM1* in the RPE/choroid. We found the DSG for *TRPM1* to be significantly down regulated in the macular retina in NEO vs. normal and significantly upregulated in iAMD vs. NEO (Table 6). However, when examining our DEGs, *TRPM1* was significantly upregulated in iAMD vs. NEO in macula RPE/choroid (Table 4).

The allele-specific expression (ASE) of *CFH* demonstrated significant allelic imbalance in both iAMD and NEO depending on the SNP being interrogated (Table 8), although *CFH* did not demonstrate differential gene expression between disease states. It may be this unequal expression of alleles at a given variant within the *CFH* gene that contributes to the disease pathophysiology of AMD. The mechanisms that underlie ASE are under active investigation and include epigenetics [15]. The evaluation of known coding regions in previously reported GWAS loci demonstrated that significant ASE for *C3*, rs2230199, and *CFH*, rs1061170, occurred in the macular RPE/choroid for normal and iAMD, while ASE for *CFH*, rs1061147, occurred in the macular RPE/choroid for normal and intermediate and neovascular AMD (Table 8). The protective variant for *CFB*, rs641153, only demonstrated ASE in the normal macular RPE/choroid (Table 8). Findings regarding the complement genes *C3* and *CFH* suggest that coding variants at these loci may influence AMD development via an imbalance in gene expression in a tissue specific manner. A similar circumstance has been noted for the inverse pattern of association of the *APOE* alleles; the ε4 allele increases the risk of AD and the ε2 allele is protective, whereas the effects of these alleles on AMD risk are the opposite [7,113,114,115,116]. Interestingly, the FDA has approved two inhibitors of complement, pegcetacoplan (C3) and avacincaptad pegol (C5), as the first medications to treat geographic atrophy (dry AMD).

In summary, this RNA-Seq experiment identified novel DEGs/DSGs that may be acting in concert, along other with factors such as ASE and miRNAs, contributing to the development of intermediate and neovascular AMD. It also expanded upon previous gene expression studies that demonstrated differential gene expression in affected tissues. Our results may provide insight into why some, but not all individuals with intermediate AMD develop advanced forms of the disease. Gene expression, along with splicing, may help to refine the pool of candidates for further investigation for therapeutic targets.

## Figures and Tables

**Figure 1 cells-12-02668-f001:**
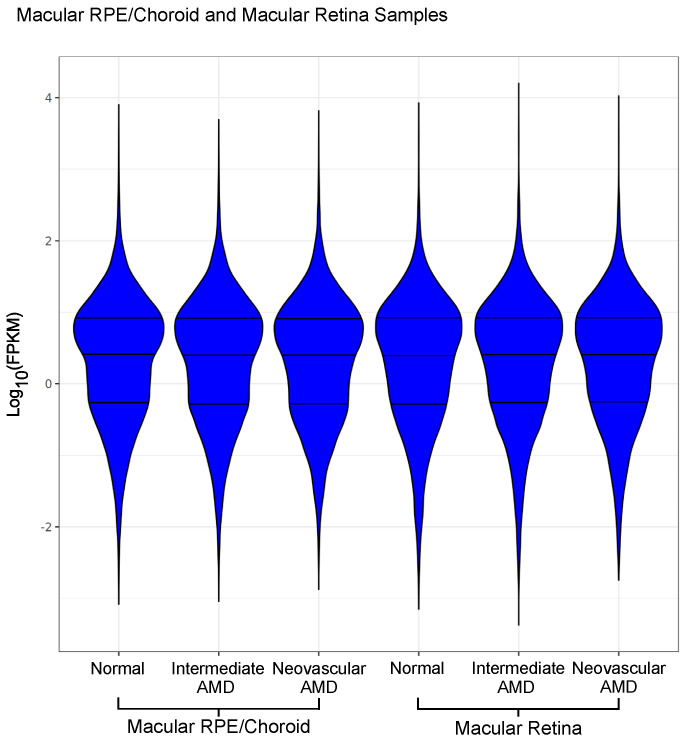
Violin plot of Log_10_-transformed FPKM counts from 27 donor eye samples with both the macular RPE/choroid and macular retina shown. Abbreviations: AMD, age-related macular degeneration, RPE, retinal pigment epithelium, FPKM, fragments per kilobase of transcript per million mapped reads.

**Figure 2 cells-12-02668-f002:**
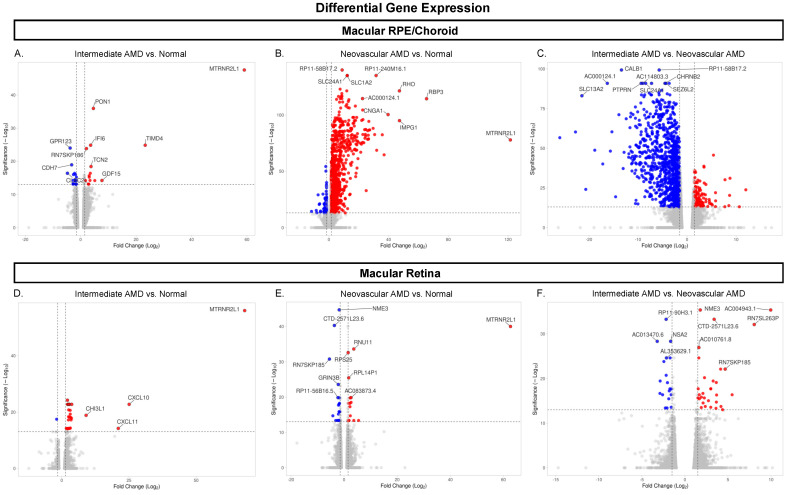
Volcano plots of differentially expressed genes across disease states. (**A**–**F**) Each dot represents one of the 26,650 genes expressed. Blue and red represent significant genes, with red indicating upregulation and blue indicating downregulation in each disease comparison. Grey dots represent genes that did not meet the significance threshold of *padj* < 0.05 and a fold change ≥ |1.5|. The ten most significant genes in each disease comparison are labeled. Abbreviations: AMD, age-related macular degeneration, RPE, retinal pigment epithelium.

**Figure 3 cells-12-02668-f003:**
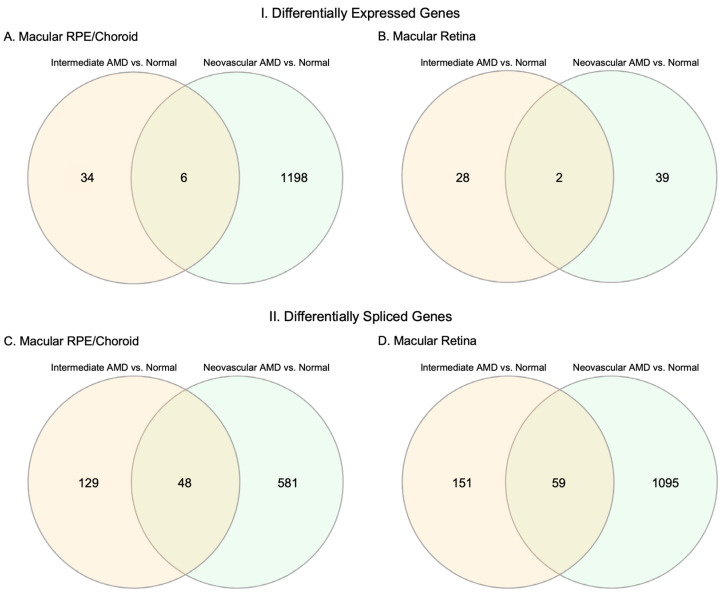
Overlap of differentially expressed genes (DEGs) and differentially spliced genes (DSGs) between intermediate AMD (iAMD) vs. normal and neovascular AMD (NEO) vs. normal. (**A**–**D**) Each circle represents the number of significant DEGs or DSGs in macular RPE (retinal pigment epithelium)/choroid and macular retina. The overlap between these two circles shows the number of overlapping genes that were regulated in the same direction between each comparison. Abbreviations: AMD, age-related macular degeneration, RPE, retinal pigment epithelium.

**Figure 4 cells-12-02668-f004:**
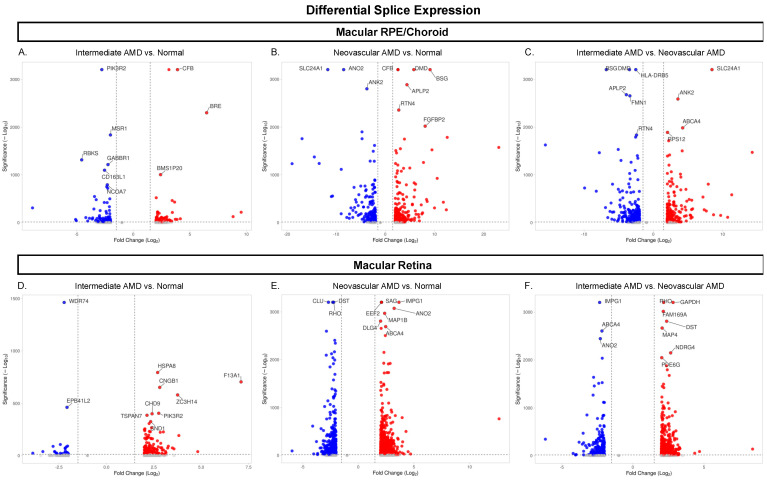
Volcano plots of differentially spliced genes across disease states. (**A**–**F**) Each dot represents one of the 26,650 genes expressed. Blue and red represent significant genes, with red representing upregulation and blue representing downregulation in each disease comparison. Grey dots represent genes that did not meet the significance threshold of *padj* < 0.05 and a fold change ≥ |1.5|. The ten most significant genes in each disease comparison are labelled. Abbreviations: AMD, age-related macular degeneration, RPE, retinal pigment epithelium.

**Figure 5 cells-12-02668-f005:**
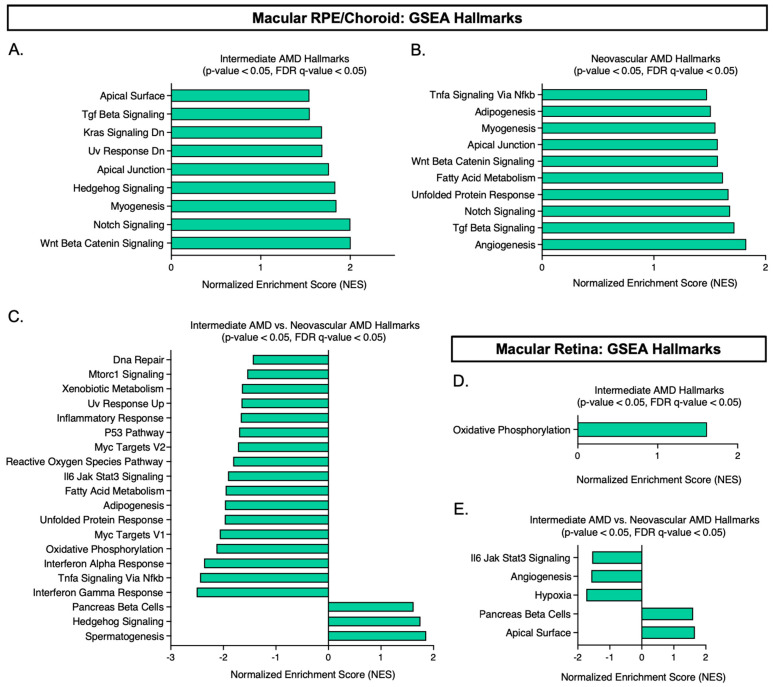
Gene set enrichment analysis (GSEA) using our normalized expression dataset. Thresholds were set based on the nominal *p*-value < 0.05 and FDR q-value ≤ 0.05 generated by the GSEA software, v.4.3.2. (**A***–***C**) show the significant hallmarks (*p*-value < 0.05, FDR q-value ≤ 0.05) identified in the macular RPE/choroid across disease state. (**D**,**E**) show the significant hallmarks (*p*-value < 0.05, FDR q-value ≤ 0.05) identified in the macular retina across disease state. Please note: no hallmark was identified to be significantly upregulated in neovascular AMD for the macular retina. Abbreviations: GSEA, gene set enrichment analysis, AMD, age-related macular degeneration, RPE, retinal pigment epithelium.

**Figure 6 cells-12-02668-f006:**
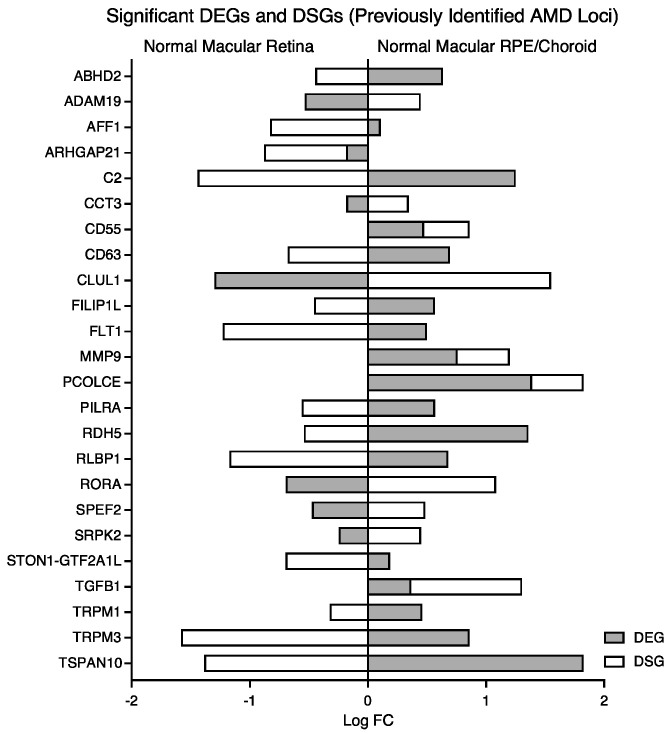
Visualization of significant DEGs and DSGs (previously associated with AMD) found in normal macular RPE/choroid vs. normal macular retina to illustrate directionality in normal tissues. Abbreviations: AMD, age-related macular degeneration, RPE, retinal pigment epithelium, DEG, differentially expressed gene, DSG, differentially spliced gene.

**Figure 7 cells-12-02668-f007:**
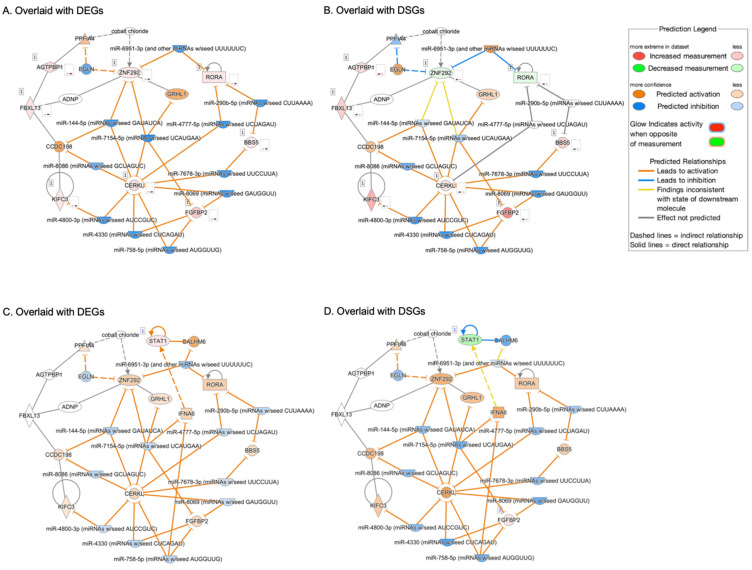
Ingenuity Pathway Analysis (IPA)-generated network of our validated genes (DEGs, DSGs, and confirmed in an independent bulk RNAseq dataset): (**A**,**B**) show the 7 identified genes from NEO vs. normal; (**C**,**D**) show our 7 identified genes from NEO vs. normal combined with *STAT1*, our validated gene from iAMD vs. normal. Each network is overlaid with expression values/fold changes from either our DEG or DSG dataset for that disease state comparison. Red or green indicates the gene was found in our dataset and associated with increased or decreased measurement, respectively. Orange or blue indicates the gene was not in our dataset but is predicted to be associated with activation or inhibition. Further clarification is provided in the legend. Abbreviations: DEG, differentially expressed gene, DSG, differentially spliced gene.

**Table 1 cells-12-02668-t001:** Subject characteristics of the bulk RNAseq discovery dataset.

**Normal**					
**Group**	**N**	**Avg. RIN**	**Age (Range)**	**Males**	**Females**
Macular RPE/Choroid (All Samples)	12	6.66	74.0 (60–94)	9	3
Macular RPE/Choroid (Outliers Removed)	9	6.93	74.2 (60–94)	7	2
Macular Retina (All Samples)	12	6.65	74.0 (60–94)	9	3
Macular Retina (Outliers Removed)	10	6.76	74.4 (60–94)	8	2
**Intermediate AMD**					
**Group**	**N**	**Avg. RIN**	**Age (Range)**	**Males**	**Females**
Macular RPE/Choroid (All Samples)	10	6.70	76.0 (60–87)	6	4
Macular RPE/Choroid (Outliers Removed)	9	6.76	75.0 (60–87)	7	2
Macular Retina (All Samples)	10	6.89	76.0 (60–87)	6	4
Macular Retina (Outliers Removed)	9	6.91	75.0 (60–87)	6	3
**Neovascular AMD**					
**Group**	**N**	**Avg. RIN**	**Age (Range)**	**Males**	**Females**
Macular RPE/Choroid (All Samples)	5	7.06	83.4 (74–94)	2	3
Macular RPE/Choroid (Outliers Removed)	5	7.06	83.4 (74–94)	2	3
Macular Retina (All Samples)	5	6.70	83.4 (74–94)	2	3
Macular Retina (Outliers Removed)	5	6.70	83.4 (74–94)	2	3

Abbreviations: N, number; Avg., average; RIN, RNA integrity number; RPE, retinal pigment epithelium.

**Table 2 cells-12-02668-t002:** Comparison of previously identified AMD genes to differentially expressed genes (DEGs) in macular RPE/Choroid vs. macular retina of normal tissue. A “+” sign indicates the gene was upregulated, while a “−” indicates the gene was downregulated.

Normal Macular RPE/Choroid vs. Normal Macular Retina							
AMD Associated Loci	*Padj*-Value DEG	Fold Change DEG	RPE/Retina DEG	AMD Associated Loci	*Padj*-Value DEG	Fold Change DEG	RPE/Retina DEG
ABCA1	1.39 × 10^−83^	+13.86	RPE	LRP6	1.68 × 10^−19^	+2.71	RPE
ABCA7	6.44 × 10^−13^	−6.37	Retina	ME3	2.88 × 10^−4^	+1.69	RPE
ABHD2	3.86 × 10^−21^	+4.30	RPE	MMP19	1.86 × 10^−31^	+5.40	RPE
ACAA2	1.23 × 10^−16^	+2.54	RPE	MMP9	4.07 × 10^−5^	+5.72	RPE
ADAM19	9.01 × 10^−23^	−3.42	Retina	MYO1E	3.14 × 10^−79^	+9.18	RPE
ADAMTS9-AS1	2.85 × 10^−24^	+11.73	RPE	NLRP2	1.66 × 10^−3^	−4.16	Retina
ADAMTS9-AS2	1.16 × 10^−46^	+16.65	RPE	NPLOC4	3.47 × 10^−16^	+1.78	RPE
AFF1	1.21 × 10^−3^	+1.28	Retina	OCA2	2.00 × 10^−71^	+64.02	RPE
ARHGAP21	8.32 × 10^−7^	−1.53	Retina	PCOLCE	4.88 × 10^−85^	+24.56	RPE
B3GALTL	1.50 × 10^−4^	−1.36	Retina	PDGFB	7.41 × 10^−63^	+14.29	RPE
C10orf88	6.79 × 10^−35^	−2.16	Retina	PELI3	8.96 × 10^−22^	−2.51	Retina
C2	1.71 × 10^−21^	+17.82	RPE	PILRA	1.01 × 10^−6^	+3.70	RPE
C3	3.73 × 10^−24^	+16.82	RPE	PKP2	1.04 × 10^−62^	+6.14	RPE
C4A	2.28 × 10^−19^	+19.68	RPE	PLA2G4A	6.81 × 10^−33^	+6.86	RPE
C5	1.47 × 10^−9^	+2.24	RPE	RAD51B	7.50 × 10^−5^	+1.63	RPE
C9	6.47 × 10^−31^	+14.42	RPE	RASIP1	1.34 × 10^−94^	+14.43	RPE
CCT3	6.66 × 10^−6^	−1.52	Retina	RDH5	4.47 × 10^−34^	+22.82	RPE
CD46	1.07 × 10^−9^	+2.02	RPE	RGS13	1.40 × 10^−7^	+9.17	RPE
CD55	8.99 × 10^−22^	+2.98	RPE	RLBP1	1.63 × 10^−9^	+4.80	RPE
CD63	3.07 × 10^−65^	+4.95	RPE	ROBO1	2.80 × 10^−7^	+1.72	RPE
CDH7	2.73 × 10^−231^	−83.36	Retina	RORA	1.19 × 10^−30^	−4.95	Retina
CDH9	2.36 × 10^−17^	−31.33	Retina	RORB	3.68 × 10^−23^	−3.03	Retina
CETP	1.97 × 10^−35^	+146.22	RPE	RP1L1	1.20 × 10^−24^	−37.69	Retina
CFB	7.13 × 10^−26^	+29.72	RPE	RRAS	4.74 × 10^−55^	+8.90	RPE
CFH	4.79 × 10^−179^	+62.53	RPE	SERPINA1	1.23 × 10^−13^	+14.19	RPE
CFHR3	1.56 × 10^−16^	+44.62	RPE	SKIV2L	1.72 × 10^−14^	+1.63	RPE
CFI	1.44 × 10^−12^	+3.51	RPE	SLC16A8	6.76 × 10^−54^	+54.98	RPE
CLUL1	8.25 × 10^−19^	−19.87	Retina	SMAD3	8.24 × 10^−96^	+8.51	RPE
CNN2	6.84 × 10^−60^	+8.02	RPE	SPEF2	3.60 × 10^−25^	−2.98	Retina
COL5A1	7.78 × 10^−32^	+7.65	RPE	SRPK2	2.29 × 10^−39^	−1.77	Retina
COL8A1	3.09 × 10^−106^	+90.43	RPE	STON1	6.39 × 10^−3^	+1.81	RPE
CSK	3.38 × 10^−2^	+1.45	Retina	STON1-GTF2A1L	4.51 × 10^−2^	+1.54	RPE
CYP24A1	1.12 × 10^−4^	−5.64	Retina	SYN3	1.16 × 10^−74^	−17.08	Retina
DDR1	6.88 × 10^−5^	−1.77	Retina	TGFB1	1.76 × 10^−11^	+2.32	RPE
EXOC3L2	9.79 × 10^−72^	+139.24	RPE	TGFBR1	9.59 × 10^−24^	+3.73	RPE
FILIP1L	2.63 × 10^−46^	+3.68	RPE	TIMP3	3.04 × 10^−129^	+54.19	RPE
FLT1	2.33 × 10^−14^	+3.15	RPE	TMEM97	3.66 × 10^−12^	−3.67	Retina
HERC2	2.52 × 10^−4^	−1.24	Retina	TNF	4.30 × 10^−8^	+24.29	RPE
HLA-DQB1	8.10 × 10^−11^	+15.72	RPE	TNFRSF10A	3.57 × 10^−58^	+17.43	RPE
HTRA1	1.17 × 10^−2^	−1.52	Retina	TRPM1	1.20 × 10^−12^	+2.88	RPE
IER3	5.43 × 10^−17^	+14.33	RPE	TRPM3	3.97 × 10^−40^	+7.24	RPE
IGFBP7	2.05 × 10^−279^	+31.68	RPE	TSPAN10	2.60 × 10^−80^	+67.02	RPE
IL6	3.58 × 10^−9^	+41.87	RPE	TYR	7.34 × 10^−127^	+794.49	RPE
ITGA7	1.66 × 10^−32^	+3.61	RPE	UNC93B1	1.38 × 10^−47^	+25.51	RPE
KMT2E	2.51 × 10^−11^	−1.47	Retina	VDR	1.29 × 10^−7^	+4.98	RPE
LBP	1.86 × 10^−11^	+214.07	RPE	VTN	8.35 × 10^−7^	−4.78	Retina
LIPC	2.41 × 10^−15^	+13.30	RPE	ZNF385B	4.83 × 10^−43^	−17.67	Retina

Abbreviations: AMD, age-related macular degeneration, DEG, differentially expressed gene, RPE, retinal pigment epithelium, *padj*-value, adjusted *p*-value.

**Table 3 cells-12-02668-t003:** Comparison of previously identified AMD genes: differentially spliced genes (DSGs) in macular RPE/Choroid vs. macular retina of normal tissue. A “+” sign indicates the gene was upregulated, while a “−” indicates the gene was downregulated.

Normal Macular RPE/Choroid vs. Normal Macular Retina			
AMD Associated Loci	*Padj*-Value DSG	Fold Change DSG	RPE/Retina DSG
ABHD2	1.16 × 10^−14^	−2.79	Retina
ADAM19	1.05 × 10^−2^	+2.80	RPE
AFF1	5.95 × 10^−251^	−6.74	Retina
ARHGAP21	1.06 × 10^−78^	−4.96	Retina
C2	3.65 × 10^−205^	−27.63	Retina
CCT3	2.48 × 10^−3^	+2.21	RPE
CD55	4.79 × 10^−24^	+2.43	RPE
CD63	5.40 × 10^−99^	−4.77	Retina
CLUL1	3.02 × 10^−283^	+35.49	RPE
FILIP1L	1.98 × 10^−04^	−2.85	Retina
FLT1	6.22 × 10^−38^	−16.98	Retina
GTF2A1L	2.04 × 10^−3^	+3.79	RPE
MMP9	1.10 × 10^−12^	+2.78	RPE
PCOLCE	1.30 × 10^−9^	+2.73	RPE
PILRA	3.93 × 10^−65^	−3.64	Retina
RDH5	5.27 × 10^−94^	−3.49	Retina
RLBP1	1.00 × 10^−320^	−14.86	Retina
RORA	2.48 × 10^−296^	+12.15	RPE
SPEF2	4.69 × 10^−8^	+3.05	RPE
SRPK2	1.25 × 10^−24^	+2.81	RPE
STON1-GTF2A1L	7.62 × 10^−82^	−4.98	Retina
TGFB1	1.00 × 10^−320^	+8.69	RPE
TRPM1	1.00 × 10^−320^	−2.11	Retina
TRPM3	1.00 × 10^−320^	−38.06	Retina
TSPAN10	1.00 × 10^−320^	−24.20	Retina
ZBTB38	4.97 × 10^−113^	+8.88	RPE

Abbreviations: AMD, age-related macular degeneration, DSG, differentially spliced gene, RPE, retinal pigment epithelium, *padj*-value, adjusted *p*-value.

**Table 4 cells-12-02668-t004:** Differentially expressed genes (DEGs) across disease states in the macular RPE/choroid that were previously identified as AMD risk loci. Please note: no differential gene expression for these genes was identified in macular retina comparisons. A “+” sign indicates the gene was upregulated, while a “−” indicates the gene was downregulated. An asterisk (*) represents that the gene was upregulated in the more severe disease state (i.e., intermediate AMD or neovascular AMD).

Macular RPE/Choroid: AMD Associated Loci (DEGs)								
Intermediate AMD vs. Normal			Neovascular AMD vs. Normal			Intermediate AMD vs. Neovascular AMD		
Gene Name	Fold Change	*Padj*-value	Gene Name	Fold Change	*Padj*-Value	Gene Name	Fold Change	*Padj*-Value
CDH7	−3.3	0.0128	ABCA7 *	+3.4	0.0018	ABCA7 *	−3.2	0.0032
			CLUL1 *	+15.3	1.5 × 10^−9^	CLUL1 *	−8.5	6.6 × 10^−6^
			FLT1	−1.9	0.0081	RP1L1 *	−7.0	0.0001
			RASIP1	−1.6	0.0246	SPEF2 *	−1.5	0.0165
			RORα *	+1.9	0.0043	TNFRSF10B	+1.7	0.0183
			RP1L1 *	+13.3	5.30 × 10^−8^	TRPM1	+1.7	0.0410
			VTN *	+3.3	0.0206	ZNF385B *	−4.8	7.8 × 10^−8^
			ZNF385B *	+4.3	4.3 × 10^−7^			

Abbreviations: AMD, age-related macular degeneration, DEG, differentially expressed gene, RPE, retinal pigment epithelium, *padj*-value, adjusted *p*-value.

**Table 5 cells-12-02668-t005:** Differentially spliced genes (DSGs) across disease states in the macular RPE/choroid found to be previously associated with AMD. A “+” sign indicates the gene was upregulated, while a “−” indicates the gene was downregulated. An asterisk (*) indicates that the gene is upregulated in the more severe disease state (i.e., intermediate AMD or neovascular AMD).

Macular RPE/Choroid: AMD Associated Loci (DSGs)								
Intermediate AMD vs. Normal			Neovascular AMD vs. Normal			Intermediate AMD vs. Neovascular AMD		
Gene Name	Fold Change	*Padj*-Value	Gene Name	Fold Change	*Padj*-Value	Gene Name	Fold Change	*Padj*-Value
C2	−5.0	0.000131	ABCA7	−2.0	4.9 × 10^−12^	ABHD2 *	−2.4	1.9 × 10^−12^
CFB *	+3.9	3.4 × 10^−321^	ABHD2 *	+3.9	1.5 × 10^−7^	CFB	+2.3	3.2 × 10^−121^
CLUL1	−2.1	0.0460	AFF1	−2.0	7.7 × 10^−23^	CHD9	+2.2	3.4 × 10^−15^
FLT1	+2.2	8.35 × 10^−7^	CFB *	+2.6	6.2 × 10^−321^	CLUL1	+11.3	5.1 × 10^−59^
			CLUL1	−13.5	1.2 × 10^−124^	RORα	+2.1	7.7 × 10^−10^
			RORα	−2.0	1.6 × 10^−14^	SPEF2	+2.3	0.0192

Abbreviations: AMD, age-related macular degeneration, DSG, differentially spliced gene, RPE, retinal pigment epithelium, *padj*-value, adjusted *p*-value.

**Table 6 cells-12-02668-t006:** Differentially spliced genes (DSGs) across disease states in the macular retina determined to be previously associated with AMD. A “+” sign indicates the gene was upregulated, while a “−” indicates the gene was downregulated. An asterisk (*) indicates that the gene was upregulated in the more severe disease state (i.e., intermediate AMD or neovascular AMD).

Macular Retina: AMD Associated Loci (DSGs)								
Intermediate AMD vs. Normal			Neovascular AMD vs. Normal			Intermediate AMD vs. Neovascular AMD		
Gene Name	Fold Change	*Padj*-Value	Gene Name	Fold Change	*Padj*-Value	Gene Name	Fold Change	*Padj*-Value
ACAD10	−2.2	2.1 × 10^−7^	ARHGAP21	−2.1	1.7 × 10^−44^	ADAM19	+2.1	6.3 × 10^−9^
CCT3 *	+2.7	6.1 × 10^−15^	C3 *	+2.2	0.0169	ARHGAP21	+2.2	4.82 × 10^−35^
CHD9 *	+2.4	7.4 × 10^−41^	CLUL1 *	+2.6	8.2 × 10^−139^	COL4A3	+2.1	9.0 × 10^−35^
			HERC2	−2.7	1.6 × 10^−15^	HERC2	+2.0	2.6 × 10^−11^
			LRP2	−2.1	3.8 × 10^−63^	LRP2	+2.4	1.4 × 10^−180^
			ME3 *	+2.6	0.0347	SKIV2L	+2.1	2 × 10^−22^
			ROBO1	−2.1	1.2 × 10^−20^	SMAD3 *	−2.4	5.2 × 10^−7^
			SMAD3 *	+2.4	2.4 × 10^−8^	SPEF2 *	−2.1	9.8 × 10^−5^
			SPEF2 *	+2.4	1.4 × 10^−12^	TRPM1	+2.3	1.4 × 10^−15^
			TRPM1	−2.8	2.7 × 10^−16^	ZNF385B *	−2.7	5.4 × 10^−6^
			ZNF385B *	+2.9	6.1 × 10^−10^			

Abbreviations: AMD, age-related macular degeneration, DSG, differentially spliced gene, *padj*-value, adjusted *p*-value.

**Table 7 cells-12-02668-t007:** Genes identified as DEGs/DSG and validated in an independent bulk RNAseq dataset. A “+” sign indicates the gene was upregulated, while a “−” indicates the gene was downregulated. An asterisk (*) indicates an opposing log fold change between DEG and DSG expression.

Validated Genes Across DEGs, DSGs, and a Bulk RNASeq Dataset								
**Macular RPE/Choroid: Intermediate AMD vs. Normal**								
		**Discovery DEG**		**Discovery DSG**			**Validation Bulk RNA Seq**	
Gene Name	Location hg19	Log FC	Adjusted *p*-value	Splice Coordinates hg19	Log FC	Adjusted *p*-value	Log FC	Adjusted *p*-value
STAT1 *	2q32.3	+0.45	0.0486	chr2:191829088-191829424	−0.41	6.2 × 10^−43^	+0.84	2.8 × 10^−3^
**Macular RPE/Choroid: Neovascular AMD vs. Normal**								
		**Discovery DEG**		**Discovery DSG**			**Validation Bulk RNA Seq**	
Gene Name	Location hg19	Log FC	Adjusted *p*-value	Splice Coordinates hg19	Log FC	Adjusted *p*-value	Log FC	Adjusted *p*-value
AGTPBP1	9q21.33	+0.42	2.1 × 10^−8^	chr9:88168784-88169184	+0.40	6.3 × 10^−61^	+0.39	6.7 × 10^−3^
BBS5	2q31.1	+0.21	0.0018	chr2:170374704-170374880	+0.55	5.9 × 10^−17^	−0.42	4.7 × 10^−3^
CERKL	2q31.3	+0.43	0.0012	chr2:182403824-182403984	+0.38	0.012	+0.76	1.2 × 10^−4^
FGFBP2	4p15.32	+0.58	3.7 × 10^−5^	chr4:15970850-15970932	+0.91	9.6 × 10^−203^	+0.78	1.3 × 10^−3^
KIFC3	16q21	+0.19	0.0117	chr16:57880252-57880440	+0.73	5.8 × 10^−8^	−0.59	2.0 × 10^−4^
RORA *	15q22.2	+0.27	0.0043	chr15:61333304-61333332	−0.30	1.6 × 10^−14^	+0.32	8.4 × 10^−3^
ZNF292 *	6q14.3	+0.18	0.0054	chr6:87864912-87865080	×0.32	2.9 × 10^−16^	+0.33	6.9 × 10^−3^

Abbreviations: AMD, age-related macular degeneration, DEG, differentially expressed gene, DSG, differentially spliced gene, FC, fold change.

**Table 8 cells-12-02668-t008:** Allele-specific expression (ASE) of known AMD-associated SNPs. Only individuals with more than 10 reads were counted, with significant ASE displayed (*p* < 0.05).

			Macular RPE/Choroid			Macular Retina		
SNP	Location	#Hets	Normal	Intermediate AMD	Neovascular AMD	Normal	Intermediate AMD	Neovascular AMD
CFH rs1061147	chr1:196654324	18	7/7	5/6	3/4	0/0	0/0	0/0
CFH rs1061170	chr1:196659237	18	1/7	2/6	0/4	0/0	0/0	0/0
CFH rs35292876	chr1:196706642	1	0/0	0/1	0/0	0/0	0/0	0/0
CFH rs121913059	chr1:196716375	0	0/0	0/0	0/0	0/0	0/0	0/0
PLA2G4A rs2285714	chr4:110638810	15	0/1	0/3	0/0	0/3	0/1	0/1
CFI rs141853578	chr4:110685820	0	0/0	0/0	0/0	0/0	0/0	0/0
C2 rs9332739	chr6:31903804	4	0/2	0/1	0/0	0/0	0/0	0/0
CFB rs641153	chr6:31914180	6	1/1	0/0	0/0	0/0	0/0	0/0
ARMS2 rs10490924	chr10:124214448	7	0/0	0/0	0/0	0/0	0/0	0/0
APOE rs429358	chr19:45411941	0	0/0	0/0	0/0	0/0	0/0	0/0
C3 rs147859257	chr19:6718146	0	0/0	0/0	0/0	0/0	0/0	0/0
C3 rs2230199	chr19:6718387	6	2/2	2/3	0/1	0/0	0/0	0/0
			Individuals with Significant ASE (*p* < 0.05)					

Abbreviations: AMD, age-related macular degeneration, SNP, single nucleotide polymorphism; Hets, heterozygotes; RPE, retinal pigment epithelium.

## Data Availability

The raw data reported in this study cannot be deposited in a public repository because of patient privacy reasons. To request access, email the corresponding author Margaret M. DeAngelis (mmdeange@buffalo.edu). Processed data are available in ZenodoData: https://doi.org/10.5281/zenodo.7532115, and ZenodoData: https://doi.org/10.5281/zenodo.10161540.

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
