# Peer review of "Patterns of Gene Expression, Splicing, and Allele-Specific Expression Vary among Macular Tissues and Clinical Stages of Age-Related Macular Degeneration"

_cells, 2023, doi:10.3390/cells12232668_

Round 1
Reviewer 1 Report
Comments and Suggestions for Authors
Dear Editor,
The following are my comments on manuscript ID cells-2679391. Treefa Shwani et al. performed an RNA-sequencing study on the human macula retina and RPE/choroid tissues from patients of intermediate AMD (iAMD) and neovascular AMD. The authors applied a unique combined analysis comparing both differentially expressed genes (DEGs) and differentially spliced genes (DSGs), and matched their sequencing results with previously identified AMD genes and AMD loci of GWAS results. Overall, I review this manuscript as a work with novelty. The advantage of this work also includes the quality of human samples and the solid validation of retina and RPE/choroid samples (Line 323-337). Here are my comments and suggestions that may help the authors improve this work.
Major issues:
1. In Figures 4-6 and Tables 2-3, the authors should separately analyze upregulated genes and downregulated genes. It is improper to analyze by combining upregulated genes and downregulated genes as differentially expressed genes. In Figure 4, are there genes upregulated in the comparison between iAMD vs Normal, meanwhile downregulated in the comparison between Neovascular AMD)?
Moreover, in Figure 5, did authors use all differentially expressed genes (both upregulated and downregulated) for GSEA analysis? Or should authors use upregulated genes for upregulated biological processes using GSEA analysis, and use downregulated genes for downregulated biological processes using GSEA analysis?
2. For Figures 2, 3, and 5, should authors use adjusted P value instead of P value?
3. Could authors provide more explanation about DSGs analysis in methods? In Figure 6, I do not understand how DSGs were quantified. What does Log FC mean in DSGs analysis? For instance, if the Log FC of TSPAN10 of DSGs is -1, does it mean one of the TSPAN10 mRNA splicing variants is 2-fold less than controls? How did authors quantify when a certain gene has multiple splicing variants?
4. The authors intended to compare the differential expression of RNAs to the identified AMD loci. One is mRNA expression level, and the other is single nucleotide polymorphisms (SNPs). There may be SNPs that can cause the differential expression. But I do not think it is common. Could authors provide more explanation in the discussion?
5. Line 640-641, the validation of bulk RNA sequencing using traditional qPCR is important. Why is data not shown?
Minor issues:
1. As the major findings of sequencing results contain the genes of the complement pathway, I suggest authors add clinical treatment of the complement pathway in the second paragraph of the introduction. The FDA has approved an inhibitor of complement as the first medication to treat dry AMD in 2023.
2. Please spell out “NEO” at its first appearance. Does NEO mean neovascular AMD?
3. In Table 1, does the outlier mean the samples that did not pass the QC analysis? If it is, there is supposed to be 47 samples passed QC analysis, instead of 48 samples on Line 322.
4. Given that MTRNR2L1 is a major upregulated gene in both retina and RPE/choroid samples of AMD (iAMD and neovascular AMD) compared to normal controls, the authors should discuss the background and potential association between MTRNR2L1 and AMD pathogenesis.
Reviewer 2 Report
Comments and Suggestions for Authors
It was a pleasure to read the well-written manuscript « Patterns of Gene expression, Splicing, and Allele-Specific Ex-1 pression vary Among Macular Tissues and Clinical Stages of Age-related Macular Degeneration» yb Treefa Shwani and colleagues.
1. While the molecular genetic methodology reads technically sound, I am not able to evaluate the for the outcomes most critical biostatistical work up. To roborate the methodology, a comparison of outcomes between study and partner eyes would have to be provided revealing the repeatability of gene expression patterns at least within the control group.
2. While after viewing all data setting the dys-regulation factor to 1.5 may have been well reasonable, it might roborate the findings if such were also repeated based on a statistical cutoff set to p<0.02 instead of p<0.05.
Further comments :
· The results read trustworthy, but the authors forgot to exclude the impact of co-morbidities and their treatment, which namely affects inflammatory and immunoregulatory systemic disorders and their treatment, lipid metabolism pathologies requiring treatment, diabetes mellitus and finally and most importantly, treatment for AMD in the disease groups including the in the ikntroduction discussed AREDS medication and anti-VEGF therapy. The latter well documentedly significantly impacts gene expression and signaling in neovascular AMD.
· Further comments :
· In the abstract, line 49 the expression «distinct» is as unspecific as in line 50 « tissue-specific patterning of expression, while the following sentens (underscoring…) is a shell. More specific details of their exciting findings deserve to precipitate in the abstract.
· Line 53: These findings indicate that the neuroretina is not the primarily affected tissue. As vaguely discussed in lines 666 ff the the dysregulation identified in the macular retina may well have been implicated by the abundant DEGs and DSGs in the Bruch membrane complex and be not more than a compensatory bystander phenomenon.
· Lines 87/8 : Sentence not relevant, delete.
· Line 113 : define or replace « normal AMD »
· Line 337 : replace contaminated by relevantly contaminated
· Line 425 : normalized needs to be defined
· Line 472: adjusted p to be defined
· Table 7: stat1 gene was identified in iAMD, but not in nAMD compared to normal controls. This is a relevant finding namely IPA-network generation. The strength of finding it only in iAMD has to be discussed.
· The first sentence of the discussion is a rep from the intro and has to be removed.
· In the discussion, a critical appraisal of the applieds methods and their limitations is lacking to enable the reader to estimate the strength of their findings.
Taken together, an excellent work providing important insights, though no new clues on the pathophysiological processes insulting the macular retina and Bruch’s membrtane complex during development and progression of AMD.
Reviewer 3 Report
Comments and Suggestions for Authors
This is a well planned and executed study, with results that can form the basis for future AMD research. I appreciated the focus on the macula and the separation of RPE/choroid and retina to generate more robust datasets. The paper is well-written and I have just a few minor suggestions:
1. Clarify for the readers the purpose of comparing DEGs/DSGs in normal retina vs normal choroid/RPE. That section is confusing after reading the disease comparisons and just needs an introductory sentence.
2. Explain how DEGs are quantified. What does a fold-change mean when there are often many DEGs for a single gene?
3. Please put your data on publicly available database for ease of access.
Round 2
Reviewer 2 Report
Comments and Suggestions for Authors
great work, thanks